# Using the First-Eye Back-Calculated Effective Lens Position to Improve Refractive Outcome of the Second Eye

**DOI:** 10.3390/jcm12010184

**Published:** 2022-12-26

**Authors:** Nicole Mechleb, Guillaume Debellemanière, Mathieu Gauvin, Avi Wallerstein, Alain Saad, Damien Gatinel

**Affiliations:** 1Department of Ophthalmology, Rothschild Foundation Hospital, 75019 Paris, France; 2Department of Ophthalmology and Visual Sciences, McGill University, Montréal, QC H3A 0G4, Canada; 3LASIK MD, Montreal, QC H3B 4W8, Canada

**Keywords:** biometry, IOL power calculation, second eye, effective lens position, correction factor

## Abstract

The present study is a retrospective, monocentric case series that aims to compare the second-eye IOL power calculation precision using the back-calculated lens position (LP) as a lens position predictor versus using a predetermined correction factor (CF) for thin- and thick-lens IOL calculation formulas. A set of 878 eyes from 439 patients implanted with Finevision IOLs (BVI PhysIOL, Liège, Belgium) with both operated eyes was used as a training set to create Haigis-LP and PEARL-LP formulas, using the back-calculated lens position of the contralateral eye as an effective lens position (ELP) predictor. Haigis-CF, Barrett-CF, and PEARL-CF formulas using an optimized correction factor based on the prediction error of the first eye were also designed. A different set of 1500 eyes from 1500 patients operated in the same center was used to compare the basal and enhanced formula performances. The IOL power calculation for the second eye was significantly enhanced by adapting the formulas using the back-calculated ELP of the first eye or by using a correction factor based on the prediction error of the first eye, the latter giving slightly higher precision. A decrease in the mean absolute error of 0.043D was observed between the basal PEARL and the PEARL-CF formula (*p* < 0.001). The optimal correction factor was close to 60% of the first-eye prediction error for every formula. A fixed correction factor of 60% of the postoperative refractive error of the first operated eye improves the second-eye refractive outcome better than the methods based on the first eye’s effective lens position back-calculation. A significant interocular biometric dissimilarity precludes the enhancement of the second-eye IOL power calculation according to the first-eye results.

## 1. Introduction

In the era where cataract surgery is considered one of the most performed procedures, accurate refractive results are highly awaited [1]. Consequently, cataract surgeons are required not only to perform the surgery safely and effectively but also to provide precise, predictable refractive outcomes. The latest generation intraocular lens (IOL) calculation formulas have a higher accuracy [2,3], but a minority (10 to 20%) of patients still fall out of their predicted postoperative refractive target by 0.50 diopters or more [2]. In patients requiring bilateral sequential cataract surgery, the expectations increase in the second eye, especially if the first operated eye does not achieve emmetropia [4]. Multiple studies have shown that improving the results in the second eye highly relies on adjusting the IOL power selection according to the error obtained in the first operated eye. Most patients have extremely high optical symmetry [5]: decision making in the fellow eye can then rely on the first-eye results [6,7]. Previous studies have proposed partially adjusting the IOL power to correct up to 50% of the error from the first eye [4,8]. Turnbull et al. demonstrated that adjustment coefficients for IOL selection could be either a formula-specific predicted postoperative refraction adjustment or a patient-specific IOL constant adjustment [6]. Olsen et al. used the fellow-eye postoperative month 1 anterior chamber depth (ACD) to reduce the prediction error in the operative eye [9]. Recently, a polynomial regression formula based on machine learning was developed to improve the second-eye outcomes in extreme ranges of the axial length (AL) and keratometry [10]. In all cases, it is acknowledged that the main limitation of IOL power calculation formulas is the accurate prediction of the effective lens position (LP). To our best knowledge, the back-calculated LP of the first operated eye was never used directly to enhance the IOL power calculation for the second eye. In this study, we aimed to assess the contribution of the back-calculated lens position of the operated contralateral eye as a lens position predictor for the second eye, for thin- and thick-lens IOL calculation formulas. We also compared the prediction error using a predetermined correction factor versus adapting IOL power calculation formulas to a back-calculated LP.

## 2. Materials and Methods

We conducted a retrospective data extraction and analysis, including patients who benefited from sequential bilateral cataract surgery from April 2017 to December 2019 by cataract surgeons (15 surgeons in total). All patients bilaterally implanted with Finevision IOLs (BVI PhysIOL, Liège, Belgium) were initially included. Exclusion criteria were history of refractive surgery, history of macular disease limiting postoperative visual recovery, previous glaucoma surgery, history of keratoconus or any corneal ectatic disease, presence of central corneal scar. Patients who had at least one exclusion criterion were not included in either set. Included patients were seen preoperatively and at 1 month postoperatively. At each visit, refraction by an optometrist and complete ophthalmological examination by an ophthalmologist were performed. Preoperative biometric measurements were performed using Lenstar 900 (Haag-Streit AG, Koeniz, Switzerland, EyeSuite software i8.2.2.0) and keratometry was calculated using the keratometric index of 1.3375. All cataract extraction surgeries were uncomplicated phacoemulsification with an incision of 2.2 mm.

Data were collected in accordance with the tenets of the Declaration of Helsinki, and approval was obtained from the Ethics Review Board of the Canadian Ophthalmic Research Center. The dataset was randomly divided into two samples: both eyes of 439 patients were assigned to a “training set” to develop the formulas based on back-calculated LP and to determine the most effective correction factor (CF) to apply to the formulas based on the latter. A total of 1500 eyes of 1500 patients constituted the “test set” to assess both methods’ performance. In the test set, we randomly included one eye of each patient to avoid measurement bias related to redundancy. 

Training set data were used to design two modified formulas: PEARL-LP and Haigis-LP. Postoperatively, we determined the “perfect” lens position (LP) for each patient: the value that would lead to the actual postoperative manifest refraction when entered in each IOL calculation formula. Obtained contralateral LP was then used as a lens position predictor, along with classical biometric predictors, in a thin-lens formula (Haigis) and a thick-lens formula (PEARL-DGS). This design led to defining an “a3” coefficient in the Haigis formula and an additional lens position predictor in the PEARL-DGS formula. Intereye AL and anterior curvature radius (ARC) differences were also used as lens position predictors in the latter. We could not design an LP-modified Barret universal II (BU II) formula because the formula is proprietary, thus preventing the back-calculation of the LP. We also used the training set to determine the optimal correction factor (CF) to enhance second-eye refractive prediction using Haigis, BU II, and PEARL-DGS formulas. The respective optimized CF were obtained by analyzing the change in the prediction’s mean absolute error (MAE), between the predicted refraction and the postoperative manifest refraction, with increasing CF (range 0–100%). The optimal CF for each formula was the CF value that resulted in the lowest MAE of prediction for the second eye. For example, if first eye’s implanted IOL power was +20 for a formula prediction of −0.12D and postoperative manifest refraction was −0.75D, resultant PE would be −0.63D and recommended correction would be CF × (−0.63): for a CF of 0.6, recommended correction would be 0.6 × (−0.63) = −0.378D. This value can then be added to the predicted SE of the second eye to determine the corrected predicted SE.

The prediction error (PE) was defined as the postoperative manifest refraction minus the formula’s prediction: a negative value indicated a myopic error and a positive value a hyperopic one. The mean error (ME) was defined as the mean PE for the entire dataset, and the mean absolute error (MAE) as the mean absolute value of PE for the entire dataset. All calculations and analysis were performed using Python 3.9 with the following libraries: Pandas 1.4.4, Scikit-learn 1.1.2, and SciPy 1.9.1.D’Agostino’s K-squared test performed on formulas’ mean PE consistently yielded *p* values below the 0.05 threshold, thus indicating a non-normal distribution of mean PE. Statistical comparisons between formulas’ absolute prediction errors were performed using repeated measures analysis of variance (Friedman test with Wilcoxon signed-rank post hoc analyses and Bonferroni correction). The analysis method proposed by Holladay et al. [11] was not used to compare the formula performances because mean errors were not set to zero for the test set: IOL constants were adjusted in the training set and these optimized constants were retained in the test set, to better approximate real-life settings and to highlight a potential effect on the mean error of second-eye adjustment methods. The percentage of eyes with a PE within 0.25 were compared using the Cochran Q test in each subgroup. A *p* value ≤ 0.05 was considered significant.

Finally, we analyzed factors that preclude efficient second-eye enhancement. Interocular differences in AL and ARC have been previously reported as limiting factors for second-eye optimization [6]. Even though our cohort was not sufficiently large to establish a cut-off value of interpretation, we graphically represented the influence of optimization using PEARL-CF depending on interocular ARC and AL differences. We attempted to visually determine the limits of which second-eye method was no longer beneficial. Eyes were divided into subgroups of interocular difference in AL (by 0.5 mm steps) and ARC (by 0.1 mm steps). The median of the variation in error induced by applying the second-eye method was determined for each subgroup.

## 3. Results

### 3.1. Demographic Characteristics:

A total of 1939 patients (3878 paired eyes) were included. A total of 878 eyes from 439 patients were selected to constitute the training set, and 1500 eyes from 1500 patients were discarded from the remaining eyes to avoid using both eyes in the test set, leaving 1500 eyes from 1500 patients. The latter were randomly selected to compare the prediction precision of unmodified base formulas, optimized formulas according to the LP (Haigis-LP and PEARL-LP) and the fixed correction factor CF (Haigis-CF, PEARL-CF, and BUII-CF). The characteristics of the patient’s population are shown in Table 1 for both the test sets and training sets. There was no statistically significant difference between both sets. The biometric parameters in both subgroups were similar to what was previously described in the Caucasian population [12].

### 3.2. IOL Constant Optimization and Formula Design

A triple optimization was conducted as described by Haigis: the ELP was retrospectively calculated for each eye of the training set and a linear regression was applied to the ACD and AL to predict this value. The resultant values are one intercept (a0) and two coefficients (a1 applied to the ACD and a2 applied to the AL values) [13]. In our study, Haigis’ formula triple-optimized constants were a0 = −0.366, a1 = 0.399, and a2 = 0.172. A quadruple-optimized Haigis formula (Haigis-LP) was designed, using the back-calculated lens position of the contralateral eye as an additional ELP predictor (a3). Haigis’ formula quadruple-optimized constants were a0= −0.247, a1= 0.153, a2= 0.071, and a3 = 0.614. The PEARL-LP formula used the back-calculated lens position of the contralateral eye as well as intereye differences in the AL and ARC as a lens position predictor. The adjusted A-constants were 118.9127 and 119.0395 for the PEARL-DGS and BU II formulas, respectively. The optimal CF was 65% for Haigis, 58% for Barrett U II, and 53% for the PEARL-DGS formula (Figure 1).

### 3.3. Second-Eye Enhancement Outcomes

Applying the enhanced Haigis formula using the LP or CF reduced the mean absolute error of the second eye by 0.066D and 0.071D, from 0.302D to 0.236D and 0.231D, respectively. The enhanced PEARL-DGS formula using the LP or CF reduced the MAE of the second eye by 0.040D and 0.043D, from 0.262D to 0.222D and 0.219D, respectively. Finally, optimizing the BU II using a prefixed CF for the second eye reduced the MAE by 0.058D, from 0.284D to 0.226D. The reduction in the MAE with the second-eye method was statistically significant (*p* < 0.001) for the LP-adjusted formulas and the CF adjustment in comparison to all three base formulas (the Wilcoxon signed-rank test). The Haigis-CF performed significantly better than the Haigis-LP (*p* < 0.001) while no statistically significant difference was observed between the PEARL-CF and PEARL-LP. Figure 2 shows the frequency of the occurrence of prediction errors of ±0.25D, ±0.50D, ±0.75D, ±1.0D, and >1.0D. The prediction error was significantly lower using either of the optimization methods, with the CF formulas consistently leading to more eyes in the lowest error categories (Table 2). The base Haigis formula resulted in a prediction precision of ±0.50D in 81.8% of the cases. The second-eye optimization increased this percentage to 90.4 and 90.53% in the Haigis-LP and Haigis-CF, respectively. The base PEARL formula resulted in a prediction precision of ±0.50D in 86.93% of the cases. The second-eye optimization increased this percentage to 91.66 and 92.07% in the PEARL-LP and PEARL-CF, respectively. The second-eye optimization with a Barrett-CF of 58% increased the percentage of the precision of ±0.50D from 84.27 to 90.73%. The enhancement was particularly beneficial in decreasing the incidence of the high prediction error: a prediction error of >0.75D of the second eye decreased from 5.33% in the Haigis base formula to 2.4 and 2.27% in the Haigis-LP and Haigis-CF, respectively. It decreased from 3.4% in the PEARL-DGS base formula to 2.0 and 1.93% in the PEARL-LP and PEARL-CF, respectively, and from 4.94% in the BU II base formula to 2.2% in the Barrett-CF.

The PEARL-LP and Haigis-LP improved the prediction precision by >0.12D in 28.80 and 35.13%, respectively. The CF method increased the precision in the second eye by >0.12D in 28.60% of the patients in the PEARL-CF, 36.60% in the Haigis-CF, and 34.0% in the BU II-CF. Adjusting the LP-modified calculation to the second eye was detrimental (decreasing the prediction precision by 0.12–0.5D) in 15.47–14.60% of the patients and highly detrimental (decreasing the predictive precision by >0.5D) in 0.53–0.73% of the patients for the PEARL-LP and Haigis-LP, respectively. The CF-modified calculations to the second eye were detrimental in 12.73-14.87-14.40% of the patients and highly detrimental in 0.33-0.80-0.60% of the patients for the PEARL-CF, Haigis-CF, and BU II-CF, respectively (Table 3 and Appendix A). The graphical representation of the second-eye enhancement using the PEARL-CF suggested that the optimization benefits were lost whenever a difference in the AL > 0.5 mm or ARC > 0.18 mm occurred, and a negative influence on the prediction precision ensues (Figure 3). 

## 4. Discussion

The Haigis formula is a thin-lens formula (i.e., based on thin-lens equations, where the thickness of each lens is considered neglectable and where the principal planes are not taken into account). It uses the anterior chamber depth (ACD) and AL values as ELP predictors [13]. The PEARL formula is a thick-lens formula (i.e., the lens thicknesses and principal plane positions are considered in the optical calculations). It uses the ARC, AL, ACD, WTW, CCT, and LT values as lens position predictors [14]. The Barrett Universal II formula is also a thick-lens formula: its lens position predictors are not published. The lens constants are used to shift a formula’s prediction up or down, depending on the IOL model used and the surgeon’s specific outcomes. The Haigis formula uses three coefficients (a0, a1, and a2) while the PEARL formula and the BUII formula use, for convenience, a value popularized by the SRK/T formula (the A-constant).

Formula-specific adjustments, constant factors, and biometric measurements have been used as optimization methods for the second eye. Li et al. showed that the PEARL-DGS and BUII with formula-specific correction factors improved second-eye refractive outcomes in patients with long AL [15]. Refining the second-eye PEARL-DGS formula with a CF of 50% of the MAE of the first eye also positively impacted the refractive precision in short hyperopic eyes [16].

An effective lens position estimation is currently the primary source of the prediction error in cataract surgery [6]. Our goal was not to compare the prediction precision between formulas but to evaluate the benefits of optimizing each formula with an LP or fixed CF for the second eye. Therefore, we attempted to enhance thin-lens (Haigis) and thick-lens (PEARL-DGS) formulas for the second eye based on the back-calculated LP of the first eye, which acted as a biometric estimate for the second-eye LP. We also compared this novel method to constant factor optimization. Olsen [17], Jivrajka et al. [4], and Aristiodemou et al. [8] demonstrated that the partial adjustment of the first eye’s postoperative prediction outcome improves the second eye results even if multiple surgeons and different IOLs were considered. Turnbull and Barrett [6] found that the CF should be tailored specifically to each formula: coefficients of 0.30 (i.e., 30% of the first eye’s PE) for the Barrett Universal II, and 0.50 (i.e., 50% of the first eye’s PE) for the Hoffer Q, Holladay I, and SRK. In our cohort, the correction coefficient that resulted in the lower MAE was 58% for the BU II, 53% for the PEARL, and 65% for the Haigis formula (Figure 1).

In our study, a refractive error higher than 0.50D in the second eye was significantly reduced from 18.2% in the triple-optimized Haigis base formula to 9.6 and 9.47% in the Haigis-LP and Haigis-CF, respectively; from 13.07% in the PEARL-DGS formula to 8.33 and 7.93% in the PEARL-LP and PEARL-CF, respectively; and from 15.74% in the BU II to 9.27% in the BU II-CF. Although it was previously reported that the LP is the leading cause of a prediction error [9], our study showed that a fixed CF improves formulas’ precision for the second eye slightly better than optimizing formulas according to a back-calculated LP. We hypothesize that optimization with a CF of 60% might be more advantageous because the sources of errors other than the LP prediction may affect the prediction error. LP-adjusted formulas abolish LP-related errors, whereas the CF method compensates for global multifactorial prediction errors. 

The second-eye optimization method should not be applied to all cases. Relying on the first-eye refractive outcome to adjust the IOL power selection for the second eye assumes interocular similarity. Our study suggested that if the difference between both eyes was more than 0.5 mm in the AL or more than 0.18 mm in the ARC (which corresponds to the difference in corneal keratometry of approximately 0.75D), employing the PEARL-CF was deleterious. In these cases, unassisted base formulas would result in higher precision in the second eye. Similar results were reported in the literature in eyes with a low interocular correlation (an AL difference of more than 0.30 mm, a K-reading difference of more than 1.00D, or a preoperative ACD difference of more than 0.30 mm), where a partial adjustment in the calculation of the fellow eye would not improve the final outcome [4]. 

There are some limitations in the present study. First, our study is a monocentric retrospective study, and all the included patients benefited from the implantation of Finevision IOLs (PhysIOL, Liège, Belgium). Second, the second-eye refractive refinement is limited by the lens manufacturing tolerance, set by the International Organization for Standardization (ISO-11979-2) at ±0.30D for IOLs 15.0D or less, ±0.40D for IOLs more than 15.0D and 25.0D or less, ±0.50D for IOLs more than 25.0D and 30.0D or less, and ±1.0D for IOLs more than 30.0D [18]. Although most IOL manufacturers do not disclose their internal tolerances, reducing the labeled manufacturing tolerance has been shown to increase the refractive precision in clinical settings [19] and probably has a significant impact on the IOL formula development. Third, the present study excluded patients with prior refractive surgery and corneal scars, and our results are yet to be evaluated on such patients before generalizing with certainty. Nonetheless, CF and LP-adjusted formulas yielded similar results and benefits in long eyes (AL > 26 mm) and high keratometry (K > 46D) as “standard” eyes in our study. Finally, the dataset used in this paper is a random subset of the one used to develop the original PEARL-DGS formula (without the second-eye function); thus, a comparison of the PEARL-DGS formula with the Haigis and BU II formulas is impossible. While the PEARL-DGS lens position prediction algorithm used in this article was trained using the set previously described, it is impossible to evaluate the effect of the choices made in the PEARL-DGS formula design on our results. However, the best-performing CF determined for the PEARL formula was consistent with the results obtained for the other formulas. Hence, we believe that the PEARL enhancement analyzability is reasonable and well founded. 

## 5. Conclusions

A fixed correction factor of approximately 60% of the first eye’s prediction error performed best than modifying the IOL formulas by using the back-calculated effective lens position of the first eye as a lens position predictor. According to the first-eye results, the interocular biometric dissimilarity precludes the enhancement of the second-eye IOL power calculation. Further studies are required to assess the benefits of the CF with other types of IOLs, particularly on operated and irregular corneas.

## Figures and Tables

**Figure 1 jcm-12-00184-f001:**
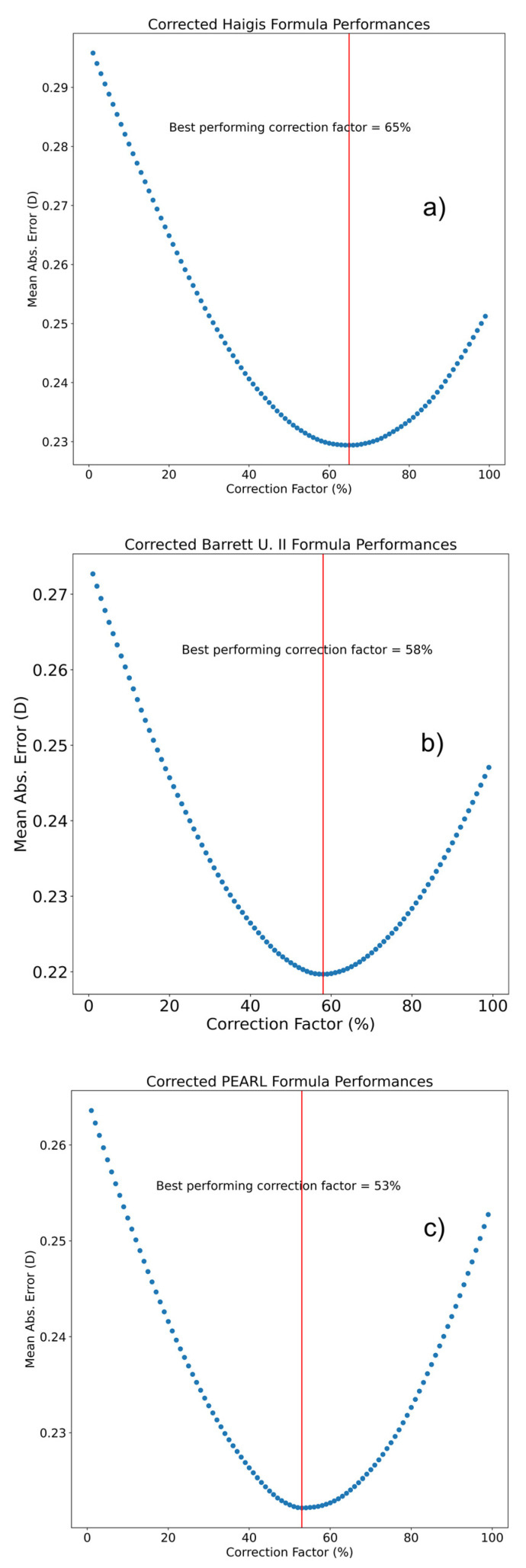
Scatter graph showing mean absolute error (MAE) of prediction in relationship to increasing correction factor (CF) in Haigis formula (**a**), Barrett universal II formula (**b**), and PEARL-DGS formula (**c**). All three formulas exhibited the lowest MAE in the second eye when a CF of around 60% (vertical red line) was applied to the first-eye prediction error.

**Figure 2 jcm-12-00184-f002:**
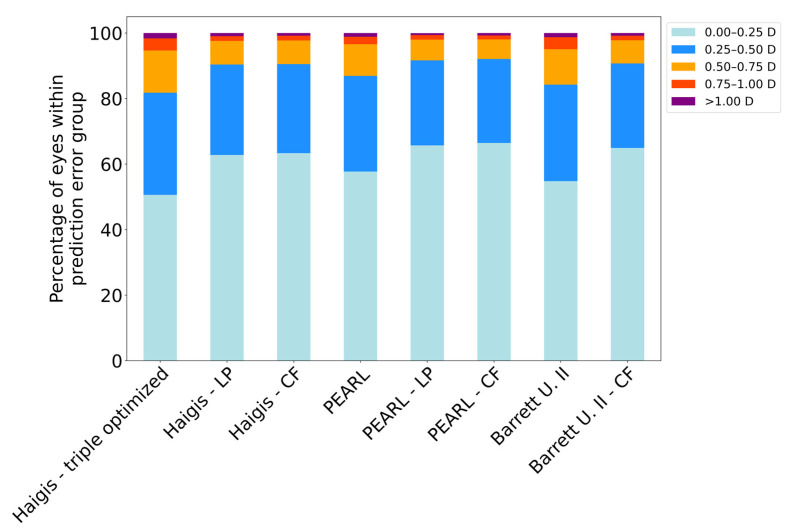
Percentage of eyes exhibiting prediction error of ±0.25D, ±0.50D, ±0.75D, ±1.0D, and >1.0D.

**Figure 3 jcm-12-00184-f003:**
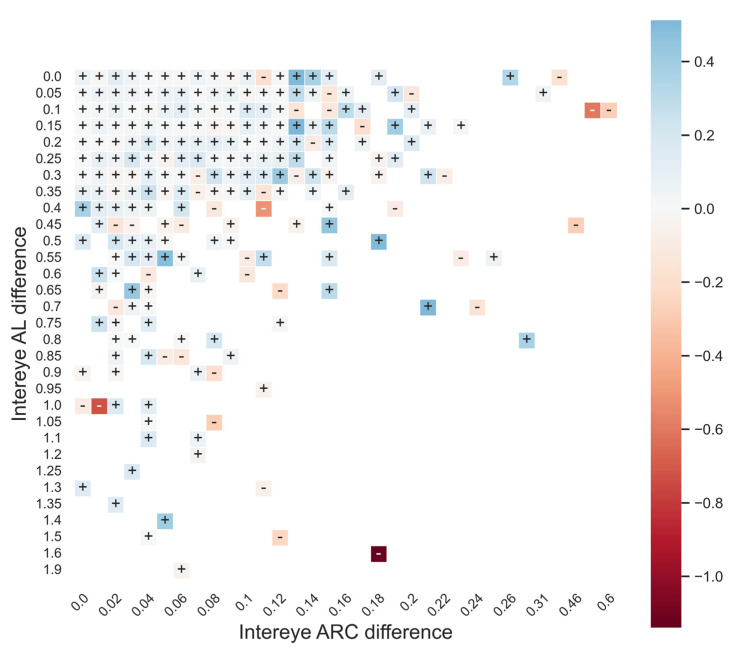
Effect of intereye difference in axial length (AL) in mm and anterior radius of curvature (ARC) in mm, on second-eye enhancement method impact on prediction error with PEARL-CF. Positive values (+) = beneficial effect, negative values (−) = detrimental effect, color scale represents the median value of prediction: blue is attributed to positive (beneficial) values and red is attributed to negative (detrimental) values.

**Table 1 jcm-12-00184-t001:** Patients Characteristics in the Training Set and the Test Set.

	Training Set (n = 878)	Test Set (n = 1500)
Female sex, n (%)	474 (54.0)	802 (53.5)
Right eye, n (%)	439 (50.0)	726 (48.4)
Age (y)	57.1 ± 6.1	57.2 ± 6.2
IOL power (D)	22.29 ± 3.0	22.27 ± 2.9
AL (mm)	23.4 ± 1.08	23.37 ± 1.07
Mean keratometry (D)	43.21 ± 1.37	43.37 ± 1.48
Anterior chamber depth (mm)	3.18 ± 0.33	3.17 ± 0.33
Lens thickness (mm)	4.4 ± 0.31	4.41 ± 0.32
Central corneal thickness (mm)	0.549 ± 0.033	0.55 ± 0.033
Corneal diameter (mm)	12.25 ± 0.4	12.23 ± 0.41
Postoperative SE (D)	−0.127 ± 0.36	−0.158 ± 0.39

Y = years, AL = axial length, D = diopter, IOL = intraocular lens, mm = millimeters, SE = spherical equivalent. Unless otherwise noted, values are mean ± standard deviation. Keratometry is calculated from the anterior corneal radius using a keratometric index of 1.3375.

**Table 2 jcm-12-00184-t002:** Comparison of second-eye formula and base formula precision.

	Sd	ME	MAE	Δ MAE/Base Formula (Wilcoxon *p*-Value)	Δ MAE CF/LP (Wilcoxon *p*-Value)	Med AE	Min	Max
Haigis—Triple Optimized	0.394	0.031	0.302	-	-	0.244	−1.557	2.358
Haigis—LP	0.313	0.01	0.236	−0.066 (<0.001)	-	0.186	−1.819	1.502
Haigis—CF	0.308	0.01	0.231	−0.071 (<0.001)	−0.005 (<0.001)	0.185	−1.564	1.544
Pearl	0.348	0.015	0.262	-	-	0.208	−1.58	2.132
Pearl—LP	0.294	0.028	0.222	−0.04 (<0.001)	-	0.173	−1.296	1.628
Pearl—CF	0.292	0.007	0.219	−0.43 (<0.001)	−0.003 (>0.05)	0.174	−1.269	1.574
Barrett U II	0.378	0.006	0.284	-	-	0.225	−1.745	2.235
Barrett U II—CF	0.302	−0.001	0.226	−0.058 (<0.001)	-	0.18	−1.36	1.896

SD = standard deviation, ME = mean error, MEA = mean absolute error. ΔMAE/base formula is the difference in MAE between base formula and optimized formula that represents the gain in precision obtained with optimization. ΔMAE CF/LP is the difference in MAE between CF formula and LP formula.

**Table 3 jcm-12-00184-t003:** Percentage of eyes in whom the second-eye method was highly beneficial (reduction in prediction error >0.50D), beneficial (error reduction of 0.12–0.5D), neutral (variation in prediction <0.12D), detrimental (error increase of 0.12–0.5D), and highly detrimental (error increase of >0.50D).

	PEARL-LP	PEARL-CF	Haigis-LP	Haigis-CF	Barrett U II-CF
Highly Beneficial	1.67	0.87	3	3.33	2.07
Beneficial	27.13	27.73	32.13	33.27	31.93
Neutral	55.2	58.33	49.53	47.73	51
Detrimental	15.47	12.73	14.6	14.87	14.4
Highly Detrimental	0.53	0.33	0.73	0.8	0.6

## Data Availability

Not applicable.

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
