# Peer review of "Using the First-Eye Back-Calculated Effective Lens Position to Improve Refractive Outcome of the Second Eye"

_jcm, 2022, doi:10.3390/jcm12010184_

Round 1
Reviewer 1 Report
This is a very interesting paper that tackles an important question since there is still room for improvement in IOL power calculation. The authors present a novel way to optimize the calculation for the second eye based on the results obtained from the first eye which is justified since, most of the times, there is a high correlation between both eyes of a patient. The authors report that their method, using back-calculation of the first eye to estimate the lens position in the second, improves the refractive results on the second eye operated in cataract surgery and even a better result is obtained using a fixed CF. Overall is an interesting paper that needs some work to better explain some sections.
The introduction is short, concise and properly explains the problem.
The manuscript is well written, but some parts are quite dens and the reader needs to re-read. Specifically, the methods section is a bit poor and it could be improved with additional information and proper explanation of concepts and methodology.
Results are clear and concise.
The discussion could be expanded.
See comments below:
2. Materials and methods
I would recommend a statistic subsection to better explain the methods. Here they can explain the MAE, the PE, the statistical test used and the significance level (also statistical power), how the calculations were done (normally you want to specify software)?
Missing exclusion/inclusion criteria. Any of these patients have had previous refractive surgery? Which is the percentage of data entries that were not valid and why? Did the authors remove data entries due to missing information, outliers…
It is a bit confusing the total number of patients. My understanding is that the total dataset is 1939 cases and it was randomly divided into two groups: group 1 contained 439 cases (22.6 %) and both eyes were used as the training set to estimate the back-calculated LP. Group 2 contained 1500 cases and one eye (the first operated eye) was used as the test set to assess both methods performances. If that is correct, could the authors better explain?
How the predetermined CF is calculated? Is this formula specific and given by the standardized IOL power calculation formulae? How is the back-calculation done? Which were the dependent and independent variables?
It would be helpful to have some background on Haigis and PEARL formulae, at least which are the variables for IOL calculation, what type of corneal measurements and biometric measures do they consider? Which are the main differences between these two methods? Especially since the authors talk about Haigis coefficients and variables in the results. If the reader is not very familiar with IOL power calculation formulae this can get confusing. Also explain what is the A-constant briefly.
Which is the procedure of the pre-operative, surgery and post-operative procedures. Which are the instruments used for the biometric, refractive and corneal measurements? Are all the cases assessed pre-operatively with the same instruments and conditions? Which are the time frames between the surgery and post-operative measurements? How many surgeons were involved? Which were the IOLs (it is said in the discussion, but it should be in methods)?
I guess the CF is for the IOL power. It seems a percentage of the formula derived power, is that CF percentage added to the calculated power? Or is it a percentage of the MAE?
Minor comments:
Line 83: the “resultant post-operative refraction” refers to the actual manifest refraction after the surgery?
Line 88: The authors mention the BU II without previously saying that is the Barret formula. Also, I would rather say that the formula is proprietary instead not published as that could confuse the lector.
Line 92: would be nice to specify that the mean absolute error is between the predicted refraction and the manifest (?) refraction after the surgery. Do the authors use the spherical equivalent? This should be outlined in the methods section.
Lines 98-100: Can the authors clarify? They say CF was not performed but in the previous line it looks like it was.
Line113: ARC has not been defined before.
3. Results
Line 121: For the test set how/which was the chosen eye?
Table 1 shows the SE that has not been described before. The fact that keratometry is calculated using the KI should be stated in the methods. Which instrument the authors used to calculate the keratometry? If the instrument allows the assessment of the total power, I think this should be used instead. It has been shown that there is a significant improvement when the total power (calculated using both anterior and posterior corneal surfaces) is used. So would be interesting to see if the LP and CF modifications also produce such an improvement when the net corneal power is used.
Line 138: how the coefficient a3 is added to the formula?
Figure 1: Please put units in the axes (% and D).
Table 2: describe med AE in the legend and specify min and max error. This is the first time “Haigis triple-optimized” appears in the text. Please define it.
Figure 3: Why the performance of the formulae depending on AL and ARC is only assessed for the PEARL-CF?
4. Methods
Line 207: AL has not been defined previously in the manuscript.
Although the demographics show that most of the eyes are in a range which is considered “normal” in terms of AL it would be interesting to now if in extremely long or short eyes the fixed CF is also performing better than the LP optimized method. And same for irregular/extreme corneal shapes. Specifically, I would like to see if for irregular corneas an optimized LP using the total cornea power yields to better results, as in these corneas it has been shown that using the keratometric index instead of the real power of the posterior corneal surface is advantageous. This should also be discussed.
This manuscript suggests, against what it would be expected, that an accurate lens position calculation is not that important, but the results are biased. Can the authors discuss this?
Which is the percentage of subjects of the dataset that were used to develop the PEARL formula. If it is a high percentage the authors should not assess this formula, if it is a reasonable percentage, could the authors do a second analysis only including the eyes that were not used in the PEARL to ascertain its outcomes compared to the other two formulas?
I think the LP optimized can be a promising method and it would be nice to see how it performs in other populations. I would say that it should beat the CF method in post-refractive patients, keratoconic or patients with corneal irregularities.
5. Conclusions
I would also say that further studies should assess if that is true for eyes with irregular corneas.
Author Response
This is a very interesting paper that tackles an important question since there is still room for improvement in IOL power calculation. The authors present a novel way to optimize the calculation for the second eye based on the results obtained from the first eye which is justified since, most of the times, there is a high correlation between both eyes of a patient. The authors report that their method, using back-calculation of the first eye to estimate the lens position in the second, improves the refractive results on the second eye operated in cataract surgery and even a better result is obtained using a fixed CF. Overall is an interesting paper that needs some work to better explain some sections.
The introduction is short, concise and properly explains the problem.
The manuscript is well written, but some parts are quite dens and the reader needs to re-read. Specifically, the methods section is a bit poor and it could be improved with additional information and proper explanation of concepts and methodology.
Results are clear and concise.
The discussion could be expanded.
See comments below:
- Materials and methods
- I would recommend a statistic subsection to better explain the methods. Here they can explain the MAE, the PE, the statistical test used and the significance level (also statistical power), how the calculations were done (normally you want to specify software)?
Answer:
Thank you for your valuable input, we added the following paragraph to the methods (line 101).
Statistics :
The prediction error (PE) was defined as the post-operative manifest refraction minus the formula’s prediction : a negative value indicated a myopic error and a positive value a hyperopic one. The mean error (ME) was defined as the mean PE for the entire dataset, and the mean absolute error (MAE) as the mean absolute value of PE for the entire dataset. All calculations and analysis were done using Python 3.9 with the following libraries : Pandas 1.4.4, Scikit-learn 1.1.2, and SciPy 1.9.1. D'Agostino's K-squared test performed on formulas’ mean PE consistently yielded P values below the .05 threshold, thus indicating a non-normal distribution of mean PE. Statistical comparisons between formulas’ absolute prediction errors were performed using repeated measures analysis of variance (Friedman test with Wilcoxon signed-rank post hoc analyses and Bonferroni correction). The analysis method proposed by Holladay et al.[11] was not used to compare the formula performances because mean errors were not set to zero for the test set. The percentage of eyes with a PE within 0.25 were compared using the Cochran Q test in each subgroup. A P value ≤.05 was considered significant.
- Missing exclusion/inclusion criteria. Any of these patients have had previous refractive surgery? Which is the percentage of data entries that were not valid and why? Did the authors remove data entries due to missing information, outliers…
Answer:
Thank you for your valuable suggestion. We added the following paragraph to the methods section
All patients who underwent bilateral cataract surgery from April 2017 to December 2019 with implantation of Finevision IOLs (BVI PhysIOL, Liège, Belgium) were considered for inclusion. Exclusion criteria were history of refractive surgery, history of macular disease limiting post-operative visual recovery, previous glaucoma surgery, history of keratoconus or any corneal ectatic disease, presence of central corneal scar. Patients who had at least one exclusion criteria were not included in either set.
It is a bit confusing the total number of patients. My understanding is that the total dataset is 1939 cases and it was randomly divided into two groups: group 1 contained 439 cases (22.6 %) and both eyes were used as the training set to estimate the back-calculated LP. Group 2 contained 1500 cases and one eye (the first operated eye) was used as the test set to assess both methods performances. If that is correct, could the authors better explain?
Answer:
We agree with the reviewer that this sentence could be confusing, we corrected it accordingly.
Both eyes of 439 patients were assigned to a "training set" to develop the formulas based on back-calculated LP, and to determine the most effective correction factor (CF) to apply to the formulas based on the latter. One eye was randomly selected in each of the remaining 1500 patients to constitute the "test set" to assess both methods' performance, to avoid statistical bias.
How the predetermined CF is calculated? Is this formula specific and given by the standardized IOL power calculation formulae? How is the back-calculation done? Which were the dependent and independent variables?
Answer
Thank you. We agree the explanation was brief. We added the following paragraph to the methods section.
To determine the optimal CF, we took the postoperative PE of the first eye, which is manifest refraction minus the respective formula’s prediction, and multiplied it by a value (CF) ranging from 0-100%. This value was added to the second eye’s formula prediction to establish a corrected predicted spherical equivalent. The corrected predicted refraction was compared to manifest refraction of the second eye to determine the optimal CF.
It would be helpful to have some background on Haigis and PEARL formulae, at least which are the variables for IOL calculation, what type of corneal measurements and biometric measures do they consider? Which are the main differences between these two methods? Especially since the authors talk about Haigis coefficients and variables in the results. If the reader is not very familiar with IOL power calculation formulae this can get confusing. Also explain what is the A-constant briefly.
Answer
Thank you for your pertinent comment. We agree that background explanation is important to understand the present article. We added the following paragraph to the discussion section
The Haigis formula is a thin lens formula (i.e based on thin lens equations, where the thickness of each lens is considered neglectable and where the principal planes are not taken into account). It uses the ACD and AL values as ELP predictors. The PEARL-DGS formula is a thick lens formula (i.e the lens thicknesses and principal plane positions are considered in the optical calculations). It uses the ARC, AL, ACD, WTW, CCT and LT values as lens position predictors. The Barrett Universal II formula is also a thick lens formula: its lens position predictors are not published. Lens constants are used to shift a formula’s prediction up or down, depending on the IOL model used and, on the surgeon’s, specific outcomes. The Haigis formula uses 3 coefficients (a0, a1 and a2) while the PEARL formula and the BUII formula use, for convenience, a value popularized by the SRK/T formula (the A-constant).
Which is the procedure of the pre-operative, surgery and post-operative procedures. Which are the instruments used for the biometric, refractive and corneal measurements? Are all the cases assessed pre-operatively with the same instruments and conditions? Which are the time frames between the surgery and post-operative measurements? How many surgeons were involved? Which were the IOLs (it is said in the discussion, but it should be in methods)?
Answer
Thank you. We clarified it in the methods section.
Included patients were seen pre-operatively and at 1 to 2 months post-operatively. At each visit, refraction by an optometrist and complete ophthalmological examination by an ophthalmologist were done. Pre-operative biometric measurements were done using Lenstar 900 (Haag-Streit AG, Koeniz, Switzerland, EyeSuite software i8.2.2.0). All cataract extraction surgeries were uncomplicated phacoemulsification with an incision of 2.2mm. Fifteen surgeons were involved.
I guess the CF is for the IOL power. It seems a percentage of the formula derived power, is that CF percentage added to the calculated power? Or is it a percentage of the MAE?
Answer:
We agree with the reviewer that this point should be clarified by an example. We added the following example in the methods section.
CF is the correction factor that would be applied to the contralateral PE. For example, if the first eye’s implanted IOL power was +20D for a formula prediction’s of -0.12D and post-operative manifest refraction was -0.75D, resultant PE would be -0.63D and recommended correction would be CF x -0.63 = -0.504D. For the second eye, the IOL power to select is the one recommended by the basal formula to target a refraction equal to the formula prediction + CF.
Minor comments:
Line 83: the “resultant post-operative refraction” refers to the actual manifest refraction after the surgery?
Answer:
Thank you. We modified it in line “ post-operative manifest refraction”.
Line 88: The authors mention the BU II without previously saying that is the Barret formula. Also, I would rather say that the formula is proprietary instead not published as that could confuse the lector.
Answer:
Thank you for your pertinent comment. It was modified accordingly. Line 104: We could not design a LP-modified Barret universal II (BU II) II formula since the formula is proprietary
Line 92: would be nice to specify that the mean absolute error is between the predicted refraction and the manifest (?) refraction after the surgery. Do the authors use the spherical equivalent? This should be outlined in the methods section.
Answer:
Thank you. We specified it in the text.
Line 122: “The mean error (ME) was defined as the mean PE for the entire dataset, and the mean absolute error (MAE) as the mean absolute value of PE for the entire dataset.”
Lines 98-100: Can the authors clarify? They say CF was not performed but in the previous line it looks like it was.
Answer:
We’re sorry we were not clear on that issue. We clarified the following in the As the reviewer knows, IOL constant adjustment is a mean of standardizing cohort’s results to insure comparability of IOL power calculation formulas. In our study, this adjustment was done in the training set to determine a CF and a modified LP formula. In the test set, we aimed to assess benefits in real life settings, where the surgeon would use his predetermined adjusted constant to perform the calculations : hence, A-constant adjustment was not done on the test set to highlight potential bias induced by the use of the evaluated second-eye adjustments.
Line113: ARC has not been defined before.
Answer:
Thank you. We defined it in line 138 “axial radius of curvature (ARC)”
- Results
Line 121: For the test set how/which was the chosen eye?
Answer:
We agree with the reviewer it should be clearly stated. They were randomly selected. We specified it line 125 (“randomly selected”)
Table 1 shows the SE that has not been described before. The fact that keratometry is calculated using the KI should be stated in the methods. Which instrument the authors used to calculate the keratometry? If the instrument allows the assessment of the total power, I think this should be used instead. It has been shown that there is a significant improvement when the total power (calculated using both anterior and posterior corneal surfaces) is used. So would be interesting to see if the LP and CF modifications also produce such an improvement when the net corneal power is used.
Answer
Thank you for your pertinent comment. We added the following sentence to the methods section “ keratometry displayed by the biometer was calculated using a keratometric index of 1.3375.”
We agree it would be interesting to assess total power: however, the device we used (Lenstar 900) does not measure total keratometry, and relies on the anterior corneal radius measurement and KI to determine the displayed keratometry.
Line 138: how the coefficient a3 is added to the formula?
Answer
A3 was added to the formula as an additional variable:
Classic Haigis formula is d = a0 +a1*ACD+ a2*AL
Second eye Haigis formula is : d = a0 +a1*ACD+ a2*AL + a3*d_controlateral
We clarified in it line 104 of the methods sections
Figure 1: Please put units in the axes (% and D).
Answer
Thank you for your comment, we added the units.
Table 2: describe med AE in the legend and specify min and max error. This is the first time “Haigis triple-optimized” appears in the text. Please define it.
Answer
Thank you, we agree it should be defined. We added the following paragraph to the methods section line 193.
Triple-optimization was conducted as described by Haigis: the ELP was retrospectively calculated for each eye of the training set and a linear regression was applied to ACD and AL to predict this value. The resultant values are one intercept (a0) and 2 coefficients (a1 applied to the ACD and a2 applied to AL values)[13].
Figure 3: Why the performance of the formulae depending on AL and ARC is only assessed for the PEARL-CF?
Answer
We agree this is a valid question. We only assessed the formulae depending on AL and ARC for PEARL-CF because in our cohort, it was the formula that yielded the lowest PE, which means, if extreme values decrease this particular formula’s sensitivity, they would most likely lead to the same results (if not majored) with other formulas. For the sake of brevity and to avoid excessive number of figures we opted for PEARL – CF. However, if the reviewer wishes we can add other formulas’ figures as supplementary files.
- Methods
Line 207: AL has not been defined previously in the manuscript.
Answer
Thank you, it was defined “axial length”.
Although the demographics show that most of the eyes are in a range which is considered “normal” in terms of AL it would be interesting to now if in extremely long or short eyes the fixed CF is also performing better than the LP optimized method. And same for irregular/extreme corneal shapes. Specifically, I would like to see if for irregular corneas an optimized LP using the total cornea power yields to better results, as in these corneas it has been shown that using the keratometric index instead of the real power of the posterior corneal surface is advantageous. This should also be discussed.
Answer
Thank you. This is a very interesting point since IOL power calculations are the hardest in extreme AL and atypical cases. We added the following paragraph in the discussion method.
Third, the present study excluded patients with prior refractive surgery and corneal scars, and our results are yet to be evaluated on such patients before generalizing with certainty. Nonetheless, CF and LP-adjusted formulas yielded similar beneficial results in extreme long eyes (AL>26mm) as “standard” eyes in our study.
Also, please find below a table comparing Second Eye Formula and base formula precision for eyes with AL > 26mm. If the reviewer wishes, we can it as supplementary file.
This manuscript suggests, against what it would be expected, that an accurate lens position calculation is not that important, but the results are biased. Can the authors discuss this?
Answer
We may have not been explicit about this issue. Thank you for your attentiveness. We modified it in the discussion section,
Our manuscript indeed suggests that accurate lens position calculation is less impactful than CF but it is not related to bias whatsoever.
We hypothesize that optimization with a CF of 80% might be more advantageous because biometric factors other than just LP still substantially affect prediction error. LP-adjusted formulas abolish LP-related errors, whereas the CF method compensates for global multifactorial prediction errors. Nevertheless, contribution of ELP in prediction precision is not to be undermimed: our cohort shows than even if ELP prediction is primordial in accurate prediction, enhancing its’ prediction is not sufficient to achieve the best outcomes.
Which is the percentage of subjects of the dataset that were used to develop the PEARL formula. If it is a high percentage the authors should not assess this formula, if it is a reasonable percentage, could the authors do a second analysis only including the eyes that were not used in the PEARL to ascertain its outcomes compared to the other two formulas?
Answer
In this specific article, the PEARL-DGS lens position prediction algorithm was trained using the eyes of the training set. Hence, we think that the results are valid enough to allow the PEARL formula to be analyzed. However, this Finevision dataset was used to devise and design the PEARL formula. The effect of those choices on calculations in this article are very hard, if not impossible, to evaluate. This was not clear, and we should have stated it more clearly. We modified it in the discussion section. We hope the reviewer will agree with us and agree that we state those limitations clearly.
I think the LP optimized can be a promising method and it would be nice to see how it performs in other populations. I would say that it should beat the CF method in post-refractive patients, keratoconic or patients with corneal irregularities.
Answer
We agree with reviewer. Thank you for your great feedback.
- Conclusions
I would also say that further studies should assess if that is true for eyes with irregular corneas.
Answer
Thank you, we modified it in the text.
Line 373 “Further studies are required to assess benefits of 80% CF with other types of IOLs, particularly on operated and irregular corneas.”
Reviewer 2 Report
This paragraph should be transferred - partly to results and partly to
discussion rather than being in methods
"One thousand, five hundred eyes belonging to 1500 patients were randomly selected 95 to compare the prediction precision of unmodified base formulas, optimized formulas ac- 96 cording to LP (Haigis-LP and PEARL-LP), and optimized formulas with fixed correction 97 factor CF (Haigis-CF, PEARL-CF, BUII-CF). The mean prediction error (PE) was adjusted 98 to zero for the training set. As constant optimization is not relevant for second eye en- 99 hancement in real-life settings, it was not performed for the test set."
The paper gives the cataract surgeons one more tool t be more precise in choosing the IOL for the second eye.
Author Response
This paragraph should be transferred - partly to results and partly to
discussion rather than being in methods
"One thousand, five hundred eyes belonging to 1500 patients were randomly selected 95 to compare the prediction precision of unmodified base formulas, optimized formulas ac- 96 cording to LP (Haigis-LP and PEARL-LP), and optimized formulas with fixed correction 97 factor CF (Haigis-CF, PEARL-CF, BUII-CF). The mean prediction error (PE) was adjusted 98 to zero for the training set. As constant optimization is not relevant for second eye en- 99 hancement in real-life settings, it was not performed for the test set."
The paper gives the cataract surgeons one more tool t be more precise in choosing the IOL for the second eye
Answer:
Thank you for your review, we moved this paragraph according to your advice, you may find it now in line 118 (results): One thousand, five hundred eyes belonging to 1500 patients were randomly selected 95 to compare the prediction precision of unmodified base formulas, optimized formulas ac- 96 cording to LP (Haigis-LP and PEARL-LP), and optimized formulas with fixed correction 97 factor CF (Haigis-CF, PEARL-CF, BUII-CF)
and 247: The mean prediction error (PE) was adjusted 98 to zero for the training set. As constant optimization is not relevant for second eye en- 99 hancement in real-life settings, it was not performed for the test set